# Copolymerization of Phthalic Anhydride with Epoxides Catalyzed by Amine-Bis(Phenolate) Chromium(III) Complexes

**DOI:** 10.3390/polym13111785

**Published:** 2021-05-28

**Authors:** Wiktor Bukowski, Agnieszka Bukowska, Aleksandra Sobota, Maciej Pytel, Karol Bester

**Affiliations:** 1Faculty of Chemistry, Rzeszow University of Technology, Powstańców Warszawy 6, 35–959 Rzeszow, Poland; wbuk@prz.edu.pl (W.B.); abuk@prz.edu.pl (A.B.); 2Doctoral School of Engineering and Technical Sciences at the Rzeszow University of Technology; 35-959 Rzeszów, Poland; asobota@prz.edu.pl; 3Faculty of Mechanical Engineering and Aeronautics, Rzeszów University of Technology, Powstańców Warszawy 12, 35–959 Rzeszów, Poland; mpytel@prz.edu.pl

**Keywords:** polyesters, ring-opening copolymerization, ROCOP, chromium(III) complexes

## Abstract

The effect of ligand structure on the catalytic activity of amine-bis(phenolate) chromium(III) complexes in the ring-opening copolymerization of phthalic anhydride and a series epoxides was studied. Eight complexes differing in the donor-pendant group (R_1_) and substituents (R_2_) in phenolate units were examined as catalysts of the model reaction between phthalic anhydride and cyclohexane oxide in toluene. They were used individually or as a part of the binary catalytic systems with nucleophilic co-catalysts. The co-catalyst was selected from the following organic bases: PPh_3_, DMAP, 1-butylimidazole, or DBU. The binary catalytic systems turned out to be more active than the complexes used individually, and DMAP proved to be the best choice as a co-catalyst. When the molar ratio of [PA]:[epoxide]:[Cr]:[DMAP] = 250:250:1:1 was applied, the most active complex (R_1_-X = CH_2_NMe_2_, R_2_ = F) allowed to copolymerize phthalic anhydride with differently substituted epoxides (cyclohexene oxide, 4-vinylcyclohexene oxide, styrene oxide, phenyl glycidyl ether, propylene oxide, butylene oxide, and epichlorohydrin) within 240 min at 110 °C. The resulting polyesters were characterized by *M_n_* up to 20.6 kg mol^−1^ and narrow dispersity, and they did not contain polyether units.

## 1. Introduction

The ring-opening copolymerization (ROCOP) of cyclic anhydrides and epoxides is a modern method of synthesis of polyesters that can provide the polymers characterized by the high molecular weight and narrow dispersity [1,2]. This irreversible reaction is characterized by a better atom economy and much better thermodynamic parameters than the classic polycondensation method. Furthermore, due to the great variety of commercially available co-monomers, the ROCOP of cyclic anhydrides and epoxides seems to be also a more universal method of polyester synthesis compared to a ring-opening polymerization (ROP) of lactones. The former can deliver both aliphatic and semi-aromatic polyesters, and the latter only the aliphatic ones [1,2]. The thermomechanical properties of ROCOP products can be tuned in the wide range by selecting the appropriate pairs of epoxides and cyclic anhydrides. As a result, polyesters with properties strictly dedicated to specific applications can be obtained [3]. 

The ROCOP reaction of cyclic anhydrides and epoxides has been known since the 1960s; however, it did not attract much attention from researchers for many years, due to the difficulties in obtaining high-molecular-weight polyesters and eliminating the side reactions of epoxide polymerization [4,5,6]. The first significant progress in this field was done only in 1985, when Inoue and Aida published an article describing the use of single-site porphyrin aluminum complexes as catalysts for the copolymerization of phthalic anhydride and propylene oxide [7]. The same authors also showed that Lewis bases used as nucleophilic co-catalysts, i.e., quaternary ammonium salts, improved sufficiently the catalytic activity of metal complexes (Lewis acids). The next noticeable progress in the development of catalytic systems for ROCOP occurred about twenty years later, in 2007, when Coates et al. reported on the use of *β*-diketiminate zinc complexes as catalysts for the copolymerization of cyclic anhydrides with epoxides [8]. The authors were the first to show that perfectly alternating aliphatic polyesters with the high molecular weight and the narrow molecular weight distribution might also be obtained in the ROCOP reactions when an appropriate designed catalyst is applied. Following this work, the increased attention has been paid by researchers to examine the metal complexes based on multidentate ligands. Among the examples described till now, the complexes of Zn(II), Al(III), Fe(III), Cr(III), Co(III), and Mn(III) with ligands bearing N- and/or O-donor atoms, such as *β*-diketiminates [8,9,10], salens [10,11,12,13,14,15,16,17,18,19,20,21,22,23,24,25,26,27,28,29,30,31,32], porphyrinates [7,10,15,27,33,34], and some others polydentate ligand types [35,36], turned out to be particularly useful in the ROCOP of cyclic anhydrides and epoxides. Most of them showed clearly higher catalytic activity and selectivity if they were used with a proper nucleophilic co-catalyst. Among all tested co-catalysts, 4-(dimethylamino)pyridine (DMAP) and bis(triphenylphosphine)iminium salts ([PPN]X) proved to be the most useful [1,2]. It was also shown that the properly designed catalytic systems based on Lewis acid–Lewis base pairs not only play a role of polymerization initiators by activating the molecules of epoxides but also influence on the rate of a propagation stage and the type of end groups in the final polyesters [1,2].

The detailed mechanistic studies performed by the groups of Duchateau, Darensbourg, and Coates showed that the ring-opening of epoxide is the stage that determines the rate of ROCOP reactions in the presence of catalytic systems composed of metal complexes and nucleophilic co-catalysts [11,13,28]. The first-order dependence on epoxide concentration indicates that either epoxide binding or opening is the turnover-limiting step. The newest experimental results supported by theoretical calculations point out that this stage occurs rather through a coordination insertion pathway than a concerted pathway proposed previously for epoxide/CO_2_ copolymerization [28]. According to the proposed mechanism, molecules of epoxide must be first activated by its coordination to metal centers and then undergo the ring-opening reaction under the influence of the attack of free carboxylate ions. Thus, to insert both epoxide and anhydride moieties into the growing polymer chain, a pair consisting of a metal complex catalyst (Lewis acid) and a nucleophilic co-catalyst (Lewis base) has to co-operate.

The studies performed by the groups of Lee [19] and Coates [31] showed that a nucleophilic co-catalyst can be a part of multidentate ligand molecules. The bifunctional catalytic systems derived from such ligands shows even higher activity in the ROCOP of cyclic anhydrides and epoxides than two-component catalytic systems. Furthermore, it was found that the difference in activity between mono- and two-component catalytic systems increases with decreasing their concentration. Unfortunately, the procedures of the synthesis of ligands with proper additional nucleophilic functionalities used to be more complex and usually included multistage transformations, making the use of such complex ligands much more expensive [19,37]. Thus, the catalytic systems composed of two components are used in the ROCOP of epoxides and anhydrides more commonly. Among them, the equimolar mixtures of salophen chromium(III) complexes and DMAP turned out to be particularly effective [11,12,15,17,30].

Many of the metal complexes that turned out to be effective as catalysts in the ROCOP reaction of cyclic anhydrides and epoxides were first examined successfully in the ROCOP reaction of epoxides and CO_2_ [1]. Thus, it seemed to be interesting to examine the usefulness of amine-bis(phenolate) chromium(III) complexes in the former reaction. To the best of our knowledge, there are no literature data on the use of amine-bis(phenolate) metal complexes as catalysts in the ROCOP reaction of epoxides and cyclic anhydride till now; however, such complexes were studied intensively as catalysts of the ROCOP reactions of epoxides and carbon dioxide by Kozak’s group [38,39,40,41,42,43,44,45].

## 2. Materials and Methods

The detailed information on all procedures of purification of reagents and solvents used in this work; the procedures of synthesis of amino-bis(phenolate) ligands and their chromium(III) complexes; the applied analytical techniques; and the results of NMR, MS, FTIR, GPC, and DSC analyses are presented in the Appendix A. † All copolymerization experiments were performed by using the procedure analogical to described in our previous work [30]. 

### General Copolymerization Procedure

Typical copolymerization experiments were performed in 2 mL vials. Each vial was dried by using a heat gun, cooled in a stream of argon, and placed in a glovebox before its use. Under an argon atmosphere, the vial was charged with 2.5 mmol phthalic anhydride, 10 μmol amine-bis(phenolate) chromium(III) complex and 10 μmol DMAP, fitted with a magnetic stir bar, and sealed up with a screw cap. Next, the vial was placed in a Schlenk tube filled with argon and unscrewed under a constant flow of argon. The tube was sealed up with a rubber septum and anhydrous toluene (0.5 mL), and CHO (2.5 mmol) were then added to the reaction mixture under an inert atmosphere, using glass syringes with needles. After the vial was screwed up again under an argon atmosphere inside the tube, it was placed in a heating block and heated to 110 °C. The reaction mixtures were stirred for fixed periods of time and then cooled immediately in an ice bath. The obtained crude products were dissolved in methylene chloride and then precipitated with excess methanol. The final products were dried under reduced pressure. GPC and ^1^H-NMR analyses were performed for all products. 

## 3. Results and Discussion

Amine-bis(phenolate) ligands can be relatively easily synthesized from phenols, primary amines, and formaldehyde in the Mannich reaction performed in organic solvent (in methanol, commonly) or water [38,41,42,45,46]. As with the salen ligands, they show the high complexing ability [46], and their structure can be simply modified by selecting the appropriate substrates. The last future enables to tune both the steric and electronic properties of the final metal complexes to modify their catalytic activity. The goal of our research was to determine the relationship between the structure of chromium(III) amine-bis(phenolate) complexes and their catalytic activity in the ROCOP reactions of epoxides with cyclic anhydrides. Hence, a series of eight amine-bis(phenolate) ligands (1–8), type [ONXO] (where X = N, O), differing with the substituents in phenolate units and R_1_-X groups, were first synthesized and then used for the synthesis of chromium(III) complexes **1a**–**8a** (Scheme 1). The applied synthesis procedures were based on ones described in References [38,41,42,45]. The structures of the final chromium(III) complexes were confirmed, using elementary analysis and spectral methods, including HRMS, according to the methodology described previously [47] (for details, see the Appendix A). 

To examine the catalytic activity of the synthesized complexes, the copolymerization of cyclohexene oxide (CHO) and phthalic anhydride (PA) (Scheme 2) was selected as a model reaction. The copolymerization experiments were performed in toluene at 110 °C, using the stoichiometric amounts of co-monomers. The reaction conditions, just as those described previously in References [11,30], were selected to give an opportunity to compare the obtained results with the data published previously. The progress of copolymerization and the content of ether linkages in the structure of the resulting polyesters were monitored by using ^1^H NMR spectroscopy. The number average molar mass and its distribution was assessed by gel chromatography, using polystyrene monodisperse standards (for further details, see the Appendix A).

### 3.1. Co-Catalyst Effect

Catalytic tests were started from performing the reaction of PA with CHO without any catalysts (Table 1, Run 1). As was expected, the ^1^H NMR analysis carried out after 60 min did not show any changes in the composition of the reaction mixture. This finding proved no reaction between PA and CHO in the absent of a catalyst under applied reaction conditions. Similar results were obtained previously for the reaction of maleic anhydride and cyclohexene oxide [48] or phthalic anhydride and limonene oxide [12]. 

Next, under the same reaction conditions, the ROCOP reaction was performed in the presence of **1a**, using the following initial molar ratio, [PA]_0_:[CHO]_0_:[Cr]_0_ = 250:250:1 (Table 1, Run 2). In this case, the ^1^H NMR analysis showed only 2% conversion of PA after 60 min, and the obtained product was an oligo(ether-ester) that was characterized by *M_n_* = 0.8 kg mol^−1^ and the content of 68 mol% ether units. The obtained results pointed to the low catalytic activity of **1a** itself [TOF = 5 h^−1^] and its low selectivity of polyester formation. The results were in agreement with those described previously for the ROCOP of PA and CHO carried out in the presence of both monometallic [12,30] and some bimetallic complexes [49,50]. 

As was mentioned in the Introduction part, to develop effective catalytic systems for the ROCOP of cyclic anhydrides and epoxides, the selection of a proper pair of a metal complex (Lewis acid) and a nucleophilic co-catalyst (Lewis base) is required. Electrophilic and nucleophilic centers, which are essential to provide the proper catalytic ability, can be located within a single molecule as well [1,2]. Both nonionic and ionic compounds can be used as co-catalysts. Amongst them, 4-(dimethylamino)pyridine (DMAP, nonionic) and bis(triphenylphosphine)imine salts (PPNX, i.e., PPNCl ionic) are the most commonly used. For instance, PPNCl is one of the most active among the known co-catalyst. Both DMAP and PPNX can catalyze itself the ROCOP of cyclic anhydride and epoxide at 110 °C or above this temperature [51]. 

Four nonionic organic bases were examined as co-catalysts in this work: PPh_3_ (triphenylphosphine), BuImd (1-butylimidazole), DBU (1,8-diazabicyclo[5.4.0]undec-7-ene), and DMAP. Each of them provided the higher PA conversion in the model ROCOP reaction than **1a** (Table 1, Runs 3–6). The lowest PA conversion was obtained in the case of PPh_3_ (TOF = 5 h^−1^) and the highest for DMAP (TOF = 35 h^−1^). Furthermore, the former provided the polymeric product characterized by the least molecular mass and the latter the one with the highest *M_n_*, respectively, 1.1 and 3.8 kg mol^−1^. Moreover, the ^1^H NMR analyses of the obtained products showed that they consist of about 60 mol% ether units. The relatively low catalytic activity and selectivity in the ROCOP of PA and CHO found for the applied organic bases under applied reaction conditions were in agreement with the data published previously [13,15,24,30,35,48].

Next, a series of ROCOP experiments were performed by using catalytic systems composed of equimolar amounts of **1a** and a basic co-catalyst (Table 1, Runs 7–10). Similarly, as it was observed by different authors previously for the binary systems based on porphyrin, salen, and [OSSO]-donor type bis(phenolate) Cr(III) complexes [11,15,24,30,35], synergetic effects could be observed for each pair **1a**/base examined as a catalytic system for the ROCOP of PA and CHO. The TONs for the binary systems, after 60 min, were clearly higher than the sums of the TONs obtained separately for **1a** and the bases mentioned above. The highest increase in TOF values was found for the mixture of **1a**/DMAP (65 h^−1^) and the lowest one for the least active system, **1a**/PPh_3_ (42 h^−1^) (Figure 1). However, the effect observed for **1a**/DMAP was 2–4 times lower than that found for the catalytic systems composed of salophen Cr(III) complexes and DMAP described in our previous work [30].

Furthermore, similarly as it was observed for porphyrin [15], salen [11,15,24,30], or [OSSO]-donor type bis(phenolate) [35] complexes, the addition of a nucleophilic co-catalyst to complex **1a** not only resulted in increasing the catalytic activity but also improved the selectivity of the model ROCOP reaction. The polyesters produced in the presence of the binary catalytic systems were characterized by lower contents of ether units than those formed in the cases when complex **1a** and nucleophilic co-catalysts were used separately (Table 1, Runs 7–10). The lowest content of ether units (~6 mol%) was found for the product of the copolymerization catalyzed by **1a**/DMAP and the highest for the reaction performed in the presence of **1a**/PPh_3_ (~14 mol%). However, the real percentage of ether segments must be somewhat lower due to the overlapping signals of protons characteristic for terminal polyester groups and the ones related to ether units [49]. The last statement seems to be supported by a decrease in the content of ether groups observed with increasing molecular weight of the resulting polymers.

An improvement in copolymerization selectivity observed in the case of the use of binary catalytic systems has been also described by Kozak’s group which studied the ROCOP of epoxides and CO_2_ in the presence of amine-bis(phenolate) chromium(III) complexes [38]. The authors found that complex **7a** used without any co-catalyst was nearly inactive catalytically in the ROCOP reaction of CHO and CO_2_ [TOF= 1 h^−1^] under applied reaction conditions. However, it formed an effective catalytic system when 0.5 eq. nonionic (DMAP) or ionic co-catalyst (PPNCl, PPNN_3_) was added. Furthermore, the authors noticed that complex **7a** itself delivered cyclic propylene carbonate as the only product, instead of polycarbonate, in the reaction of propylene oxide and CO_2_. Only the addition of nucleophilic co-catalysts (DMAP, PNNCl, PPNN_3_) to **7a** resulted in obtaining poly(propylene carbonate) (PPC) with the selectivity of 73–93% [TOF = 16–18 h^−1^] [39].

### 3.2. Ligand Structure Effect

The effect of ligand structure on the catalytic activity of amine-bis(phenolate) chromium(III) complexes in the ROCOP reaction between cyclohexene oxide and phthalic anhydride was next examined. The catalytic systems composed of complexes **1a–8a** and DMAP were studied (Table 2) under the following reaction conditions: [PA]_0_:[CHO]_0_:[Cr]_0_:[DMAP]_0_ = 250:250:1:1, 110 °C, in toluene. Comparing the values of TOFs calculated for the complexes differing in a pendant donor group, it was concluded that **4a** and **8a** with O-donor pendant groups formed with DMAP clearly more active catalytic systems than complexes **1a** and **7a** having N-donor pendant groups. The following order of TOF after 60 min were found: **4a** (R_1_-X = CH_2_OMe, TOF = 138 h^−1^) > **8a** (R_1_-X = 2-THF, TOF = 118 h^−1^) > **1a** (R_1_-X = CH_2_NMe_2_, TOF = 105 h^−1^) > **7a** (R_1_-X = 2-pyridyl, TOF = 95 h^−1^) (Table 2 Runs 2, 17, 32, and 37). The analogical relationships were observed for the experiments performed for 30, 90, 150, and 240 min as well. Based on the obtained results, it was also noted that both the exchange of an aliphatic ether pendant group (**4a**) for a tetrahydrofurfuryl one (**8a**) and an aliphatic amine group (**1a**) for a pyridyl one (**7a**) diminished the catalytic activity of amine-bis(phenolate) chromium(III) complexes. These findings result probably from the stronger donor ability of the heterocyclic groups compared to acyclic ones.

The further comparison of the results obtained for complexes **1a**, **4a**, **7a**, and **8a** (Table 2, Runs 1–5, 16–20, and 31–40) showed that the type of donor pendant groups influences not only on the activity of the examined catalytic systems but also on the value of molecular mass of the resulting polymers. Comparing the values of *M_n_* obtained for the comparable conversions of PA, it was concluded that the higher values of this parameter were noted in the case of the complexes having [ONNO]-type ligands than for the ones with [ONOO]-type ligands. For instance, the polymers characterized by *M_n_* = 12.6 kg mol^−1^ at 95% PA conversion, and *M_n_* = 9.1 kg mol^−1^ at 96% PA conversion were obtained respectively for **1a**/DMAP (R_1_-X = CH_2_NMe_2_) and **4a**/DMAP (R_1_-X = CH_2_OMe) (Table 2, Runs 5 and 19). Furthermore, when the dependence of *M_n_* vs. PA conversion was charted (Figure 2), the linear correlations between these two parameters were revealed. This finding points out that the ROCOP of CHO and PA in the presence of the binary catalytic systems occurred in well-controlled manner. Furthermore, another interesting relationship might be concluded from Figure 2, as well. The molecular mass of the resulting polymers depended on the type of donor atoms present in amine-bis(phenolate) ligands. The complexes having [ONNO]-type ligands formed polyesters with higher *M_n_* than those with [ONOO]-type ligands. The last finding pointed out that the metallic centers in the first type complexes showed the lower ability to facilitate the chain transfer reaction than the second type. The relation between the rates of chain transfer and propagation reaction is a key factor influencing the mass of the resulting polymers. However, there is not possible to eliminate completely all factors responsible for the chain transfer reaction in the ROCOP reaction of epoxides and cyclic anhydrides even under “water-free” reaction conditions. Despite water traces or phthalic acid, the zwitterions formed from DMAP and PA are present in the reaction mixture which are responsible for occurring the chain transfer reaction. It was shown previously [10,11,12] that the molecular mass of ROCOP products decreases with increasing the amount of a nucleophilic co-catalyst.

Probably due to the stronger interactions of [ONNO]-type amine-bis(phenolate) ligands with Cr(III) ions compared to [ONOO]-type ones, the former have a stronger influence on the electrophilic character of coordinated metal ions compared to the latter. Diminishing the electrophilicity of metal ions resulted probably in decreasing the rate of substitution reaction of the active ends of growing polymer chains (chain transfer reaction). The higher value of *M_n_* observed for the polymer formed in the presence of **7a**/DMAP compared to the one obtained by using **1a**/DMAP (Figure 2) seems to prove this statement additionally. Complex **7a** having 2-pyridyl moiety as a part of a pendant group has the second *N*-donor stronger than complex **1a** with a tertiary aliphatic amine group. Thus, the former interacts with a Cr(III) ion stronger than the latter. However, the effect of the type of *O*-donor group on the molecular mass of the resulting polymers was not observed (Figure 2). It can result from the low donor ability of both an aliphatic ether groups and tetrahydrofurfuryl groups. The *O*-metal bonds in transition metal complexes show much higher liability [42].

The differences in the catalytic properties of amine-bis(phenolate) chromium(III) complexes occurring depending on the structure of a pendant group (R_1_-X = CH_2_NMe_2_ or CH_2_OMe) were also described by Kozak’s group which examined the mentioned complexes as catalysts of the ROCOP of epoxides and CO_2_ [43,44]. According to the authors, the observed differences were connected with the formation of catalytically inactive forms of chromium(III) complexes, which might contain two molecules of DMAP coordinated to a metallic center or have a dimeric form, with chloride ions as bridges. The formation of dimeric forms of amine-bis(phenolate) chromium(III) complexes in the presence of DMAP were found previously by the Kozak’s group, based on the results of MALDI-TOF analysis, for the complexes with CH_2_OMe (**4a**), 2-pyridyl (**7a**), and 2-tetrahydrofuryl (**8a**) pendant groups [42]. Unfortunately, based on these findings, it is difficult to explain why complexes **4a** and **8a** having *O*-donor pendant groups show higher catalytic activity in the ROCOP reaction of CHO and PA compared to complexes **1a** and **7a** with *N*-donor pendant groups, such as CH_2_NMe_2_ and 2-pyridyl, what was observed in our study.

In the next step, the catalytic properties of the complexes having the same pendant groups and different substituents in phenolate moieties were compared (Table 2, complexes **1a–3a** and **4a–6a**). It was found that the complexes bearing MeO-substituents (**2a** and **5a**) showed lower catalytic activity compared to the appropriate complexes with *t*Bu groups (**1a** and **4a**), independently of the type of a pendant group. These findings pointed out a disadvantageous effect of the stronger donor substituents on the catalytic activity of amine-bis(phenolate) chromium(III) complexes in the ROCOP of CHO and PA. However, comparing the activity of **1a** and **4a** with the activity of **3a** and **6a**, it was found that the presence of electron acceptor substituents such as F atoms influenced advantageously on the catalytic ability of the complex with R_1_-X = CH_2_NMe_2_ but it was disadvantageous in the case of the complex with R_1_-X = CH_2_OMe. Furthermore, when TOF values for the complexes bearing the same pendant group (R_1_-X = CH_2_NMe_2_ or R_1_-X = CH_2_OMe), calculated for the 60 min reactions, were compared, the following orders of the activity were noted, respectively: **3a** (R_2_ = F, TOF = 115 h^−1^) > **1a** (R_2_ = *t*Bu, TOF = 105 h^−1^) > **2a** (R_2_ = OMe, TOF = 95 h^−1^) (Table 2, Runs 12, 2, and 7), and **4a** (R_2_ = *t*Bu, TOF = 138 h^−1^) > **5a** (R_2_ = OMe, TOF = 130 h^−1^) > **6a** (R_2_ = F, TOF = 98 h^−1^) (Table 2 Runs 17, 22, and 27). The analogical relationships were found for other reaction times.

The structure of R_2_ substituents in amine-bis(phenolate) ligands influences on the electron density on phenolic oxygen atoms which are engaged in donor-acceptor interactions with chromium(III) ions. Probably, this factor is responsible mainly for occurring some relationships between R_2_ structure and the catalytic activity of the [ONNO]-donor type amine-bis(phenolate) chromium(III) complexes with R_1_-X = CH_2_NMe_2_. The similar effect of substituents in phenolic units on the catalytic activity of metal complexes was also observed by our group previously for salophen chromium(III) complexes (other complexes with [ONNO]-type ligands) examined in the ROCOP reactions of epoxides and cyclic anhydrides [30]. It was found, for instance, that a decrease in the donor character of substituents in salicylaldehyde moieties (OMe < *t*Bu < H) influenced advantageously on the catalytic activity of the systems composed of salophen chromium(III) complexes and DMAP [30].

Similarly, as for the [ONOO]-type amine-bis(phenolate) chromium(III) complexes with R_1_-X = CH_2_OMe, the lack of general correlation between the donor/acceptor ability of substituents present in phenolic units and the catalytic activity was also observed previously by the Coates’s group for the salen cobalt(III) complexes used with [PPN]NO_3_ (co-catalyst) in the ROCOP reaction of propylene oxide and maleic anhydride [26]. The authors noted that the exchange of *t*Bu groups for OMe ones resulted in a slight decrease in TOF from 22 to 21 h^−1^. On the other side, the exchange of *t*Bu groups for chlorine or fluorine atoms resulted in an increase in TOF to 29 and 38 h^−1^, respectively. In turn, the exchange of *t*Bu for NO_2_ groups, which are strong electron acceptors, caused a relatively low decrease in TOF to 20 h^−1^). However, the same group has observed the clear dependence between the catalytic activity of metal complexes and the structure of phenolic units for the (salophen)Al(III)Cl/PPNCl catalytic systems used in the ROCOP reaction of and propylene oxide [25]. In this case, when the substituents at position 5 of salophen phenolic moieties were changed, the following changes in TOF were observed: *tBu* (88 h^−1^) < H (80 h^−1^) < F (49 h^−1^).

Our observations are in contrast to the results described previously by the Kozak’s group [41] which found that the exchange of R_2_ substituents in the amine-bis(phenolate) chromium(III) complexes having [ONNO]-type ligands with R_1_-X = 2-pyridyl from *t*Bu groups for OCH_3_ ones resulted in approximately four-fold increase in the reaction rate (*r_obs_* = 0.29 × 10^−4^ s^−1^ vs. *r_obs_* = 1.2 × 10^−4^ s^−1^) when the complexes were examined with DMAP (a co-catalyst) in the ROCOP reactions of cyclohexene oxide and CO_2_ [41]. The similar phenomenon was also observed for a series of salophen chromium(III) complexes in cyclohexene oxide/CO_2_ copolymerization [52]. Darensbourg et al. found that the catalytic activity of such complexes increased in the following: H < *t*Bu < OMe. The authors also noticed that the exchange of hydrogen atoms for OMe groups caused approximately three-fold increase in TOF, from 386 to 1096 h^−1^. In turn, in our previous work [30], it was found that the effect of substituents on the catalytic activity of salophen chromium(III) complexes in the ROCOP reaction of CHO and PA was completely opposite (OMe < tBu < H) and the introduction of OCH_3_ groups in the place of H atoms at R_2_ positions resulted in a decrease in TOF from 568 to 320 h^−1^.

The study performed for the ROCOP of CHO and PA also showed that the exchange of substituents R_2_ in the phenolic units of complexes **1a**–**6a**, similarly as the modification of a pendant group mentioned above, had a strong effect on the molecular mass of resulting products (Table 2). The relationships between *M_n_* and PA conversion presented in Figure 3 clearly show that the complexes with *t*Bu groups, **1a** (R_1_-X = CH_2_NMe_2_) and **4a** (R_1_-X = CH_2_OMe), provided the polymer products characterized by lower *M_n_* compared to related MeO-substituted complexes **2a** and **5a**. Unexpectedly, F-substituted complexes **3a** and **6a** also gave the products characterized by higher *M_n_* compared to *t*Bu-substituted complexes **1a** and **4a**. Comparing the values of *M_n_* for the reactions occurring with the complete conversion of PA (after 240 min), further differences in behavior of the studied catalytic systems could be easily noticed. The most active system, **4a**/DMAP (R_1_-X = CH_2_OMe, R_2_ = *t*Bu), provided the polymer product characterized by lower *M_n_* (10.3 kg mol^−1^) than **3a**/DMAP that was a system clearly less active catalytically. The latter produced the polyester characterized by the highest *M_n_,* equal to 17.6 kg mol^−1^ (Table 2, Runs 20 and 15). Bearing in mind that the molecular mass of products obtained in the ROCOP reactions depends on the mutual relations between the rates of polymer chain propagation and chain transfer/termination reactions, it might be only assumed that the change in the electron density on the metallic centers in the complexes has a large impact on the rate of all these reactions and *M_n_* of the resulting polymers.

The results of GPC analysis performed for the products obtained in 240-min experiments did not show the effect of the type of pendant group R_1_-X on the dispersity (*Ð*) of the molecular mass of the resulting polyesters. All polyesters that were obtained by using catalytic systems based on the complexes with R_2_= *t*Bu were characterized by *Ð* equal to about 1.2, independently of the pendant group structure (Table 2, Runs 5, 20, 35, and 40). Similarly, there were not observed differences in a mass dispersity for the complexes **2a** and **5a** with R_2_= OMe. Independently of the nature of a pendant group (R_1_-X = CH_2_-NMe_2_ or CH_2_-OMe), the value of *Ð* amounted to 1.2 (Table 2, Runs 10 and 25). However, the introduction of fluorine atoms as R_2_ substituents resulted in an increase in *Ð* value to about 1.3 both in the case of complex **3a** (R_1_-X = CH_2_-NMe_2_) and complex **6a** (R_1_-X = CH_2_-OMe) (Table 2, Runs 15 and 30). This increase may indicate a lowering the rate of the chain transfer reaction compared to the rate of propagation in the presence of complexes with electron-withdrawing groups in phenolate units (R_2_ = F), which in turn could explain producing the polymers with much higher average molar masses in these cases. The similar effect was observed by Coates and co-workers for the *para*-F-substituted salophen aluminum complexes used as catalysts in the reaction cyclic anhydrides with epoxides [25].

Based on the ^1^H NMR spectra recorded for the resulting reaction mixtures, the content of ether units in the final polymers was determined. It was found that the complexes studied in this work (**1a**–**8a**) formed highly selective catalytic systems when they were used with the equimolar amount of DMAP. The content of ether units in the product obtained after 240 min ranged from less than 1 mol% to about 3 mol% (Table 2). Furthermore, it was found that the type of a pendant donor group influenced only slightly on the selectivity of the formation of polyesters. The content of ether units was slightly lower for the products obtained in the presence of the complexes bearing *N*-donor pendant groups (**1a** and **7a**) (Table 2, Runs 5 and 35), compared to those obtained in the reactions catalyzed by the complexes with *O*-donor pendant groups (**4a** and **8a**) (Table 2, Runs 20 and 40). These findings point out that the catalytic systems based on the complexes with *O*-donor groups show better selectivity to ester group formation than the once composed of complexes having *N*-donor groups.

The exchange of R_2_ substituents from *t*Bu- to MeO- or F-groups did not affect the selectivity of polyester formation when R_1_-X was equal to CH_2_OMe (**4a**, **5a**, and **6a**). The content of ether units in the final products was then about 3 mol% (Table 2, Runs 20, 25, and 30). However, some differences in ROCOP selectivity were observed for the complexes with R_1_-X =CH_2_NMe_2_ (**1a**, **2a**, and **3a**). Clear worsening the selectivity was noted when R_2_-substituents were changed from *t*Bu groups (**1a**) to MeO ones (**2a**). In turn, the exchange of *t*Bu groups for fluorine atoms (**3a**) resulted in an improvement of the selectivity (Table 2, Runs 5, 10, and 15). The best ROCOP selectivity was found for complex **3a**, R_1_-X = CH_2_NMe_2_, that allowed, in combination with an equimolar amount of DMAP, to obtain an almost perfectly alternating poly(cyclohexene phthalate) with the content of ether units below 1 mol% (Table 2, Run 15).

### 3.3. Effect of Monomers to Catalytic System Ratio

Two additional series of catalytic experiments **3a**/DMAP were performed to study the effect of **3a**/DMAP loading on the ROCOP reaction of CHO and PA. The molar ratio of **3a**/DMAP to the monomers was increased or decreased two times compared to the reference experiments. The obtained results are presented in Table 3. Regardless of **3a**/DMAP concentration, the perfectly alternating poly(cyclohexene phthalates), contained below 1 mol% ether units and characterized by a narrow molecular weight distribution (*Ð* ~ 1.30), were obtained as a result. However, a slight increase in polyester mass dispersity could be noticed when the ratio of **3a**/DMAP to CHO and PA was decreased. Furthermore, in this case, the time required to complete the ROCOP reaction was reduced by half (to 120 min) compared to the reference experiment. In turn, when the amount of **3a**/DMAP was reduced, only 85% conversion of PA was noted after 480 min (Table 3, Runs 4, 9, and 14). Moreover, doubling the amount of **3a**/DMAP resulted in a decrease in the molecular mass of the obtained product, from 17.6 to 10.5 kg mol^−1^ (for ~100% PA conversion, Table 3, Runs 9 and 4). In turn, the polymer product characterized by *M_n_* = 22.8 kg mol^−1^ (for 85% PA conversion, Table 3, Run 14) was obtained when the amount of **3a**/DMAP to PA/CHO was decreased. In addition, the analysis of the obtained results revealed that, regardless of the amount of **3a**/DMAP used, linear correlations between PA conversion and *M_n_* were observed (Figure 4). The last findings indicate that the copolymerization proceeds in a well-controlled manner. Similar correlations were observed in our previous study [30] for the catalytic systems based on salophen chromium(III) complexes and DMAP used in the same model ROCOP reaction under the same temperature conditions. However, the reaction carried out in the presence of **3a**/DMAP occurred much slower compared to the reactions catalyzed by the examined (salophen)Cr(III)Cl/DMAP, although the catalytic systems based on complex **3a** provided the product characterized by higher molecular mass.

### 3.4. Effect of Epoxide Structure

The catalytic activity of **3a**/DMAP has been also examined in the ROCOP reactions of PA with a series of terminal epoxides (Scheme 3). The all reaction were performed in toluene, at 110 °C, using the following initial molar ratio: [PA]_0_:[epoxide]_0_:[Cr]_0_:[DMAP]_0_ = 250:250:1:1. The epoxides having both electron withdrawing groups (styrene oxide (SO), epichlorohydrin (ECH), phenyl glycidyl ether (PGE), and 4-vinylocyclohexene oxide (4-VCHO)) and electron donating ones (propylene oxide (PO) and butylene oxide (BO)) were used. The results obtained at this stage are presented in Table 4. In most cases, the nearly complete conversion of PA was observed after 240 min under the applied reaction conditions. Only for the reaction with SO, PA conversion was clearly lower and amounted to 73% after 240 min. Comparing the results obtained for the 60-min experiments, when PA conversion was incomplete for all examined epoxides, the following order of the reactivity could be concluded: ECH (TOF =225 h^−1^) > PO (TOF =155 h^−1^) > CHO (TOF = 115 h^−1^) > 4-VCHO (TOF = 98 h^−1^) > EFG (TOF =95 h^−1^)> BO (TOF =85 h^−1^) > SO (TOF = 58 h^−1^). From these findings follow clearly that the reactivity of epoxides in the ROCOP reaction with PA depends on steric effects stronger than the electron properties of substituents in the molecule of epoxide.

Based on the ^1^H NMR spectra of the products obtained in 240-min experiments, it was found that all the examined pairs of reactants delivered polyesters selectively in the presence of **3a**/DMAP (Appendix A). Most of the resulting poly(alkylene phthalates) comprised below 1 mol% ether units in their structure. Only in the case of poly(styrene phthalate) the content of ether units was found to be 2 mol%. The presence of an intensive absorption band in the range of 1724–1727 cm^−1^ corresponding to C = O valence stretching of ester groups in the FTIR spectra of the products was an additional proof of obtaining the poly(alkylene phthalates) (see Appendix A).

To obtain the information on the glass temperature (*T*_g_) of the resulting poly(alkylene phthalates), DSC analysis was also performed. The analysis provided the information how this physical parameter so important for polymer characteristics can be tuned by the proper selection of epoxide for the ROCOP reaction with PA (Table 4). For the polyesters obtained from the examined epoxides, the value of *T*_g_ changed from 47.7 °C for BO to 141.8 °C for CHO (Table 2, Runs 1 and 12). It gives the broad range of *T*_g_ value (94.1 °C) and shows that the ROCOP of PA and epoxides can deliver polyesters for various applications.

The number average molar mass values determined for the polyesters obtained from PA and different epoxides in the presence of **3a**/DMAP, based on GPC analyses, ranged from 6.4 to 20.6 kg mol^−1^. The highest value of *M_n_* was found for the copolymer of PA and 4-VCHO and the lowest for the one composed of PA and SO units (Table 3, Runs 6 and 4, respectively). The determined values of *M_n_* turned out to be clearly lower than it could be concluded taking the values of PA conversion into account. In addition, the GPC analysis revealed the bimodal distribution of polyester molecular mass (Appendix A). The high-molecular fraction had nearly twice higher *M_n_* than the low-molecular one. The attempts made to improve the purity of co-monomers and toluene did not result in a significant improvement in the molecular mass distribution of the final products. This bimodality turns out to be common for the anionic ring-opening copolymerization. The bimodal distribution of polyester molecular masses has been proved many times for the ROCOP reactions of epoxides and cyclic anhydrides [10,11,12,15,24,29,30,31,35]. At high monomer conversions, undesirable side reactions, such as chain transfer (occurring in the presence of bifunctional chain transfer agents, such as water, acid, or zwitterions), transesterification, and chain-end coupling reactions, often undergo during the ROCOP reaction of epoxides and cyclic anhydrides.

The use of catalytic systems composed of amino-bis(phenolate) chromium(III) complexes and DMAP in the ROCOP reactions of cyclic anhydrides and epoxides seems to have some advantages compared to the use of the systems based on salophen chromium(III) complexes. The polyesters produced by the former can be obtained as uncolored powders. The latter makes obtaining the uncolored products difficult (Appendix A).

## 4. Conclusions

Amine-bis(phenolate) chromium(III) complexes **1a**–**8a** having pendant group R_1_-X (CH_2_NMe_2_, 2-pyridine, CH_2_OMe, 2-tetrahydrofuryl) and differing in substituents R_2_ (*t*Bu, MeO, or F) in phenolate units were examined as effective catalysts for the model ring-opening copolymerization between phthalic anhydride and cyclohexane oxide. The complexes themselves showed the low catalytic activity; however, the strong synergetic effect was observed in the presence of stoichiometric amounts of nucleophilic co-catalysts. DMAP turned out to be the best co-catalyst from the series of four organic bases examined (PPh_3_, DBU, 1-butylimidazole, and DMAP).

The catalytic study showed that the activity of the binary systems **1a**–**8a**/DMAP depended on both the nature of substituents R_2_ and the pendant group R_1_-X. The complexes bearing *O*-donor pendant groups (CH_2_OMe and 2-tetrahydrofuryl) showed higher activity than the ones with *N*-donor groups (CH_2_NMe_2_ and 2-pyridyl). Furthermore, the complexes with 2-pyridyl and 2-tetrahydrofuryl units turned out to be less active than the proper complexes with R_1_-X equal to CH_2_OMe and CH_2_NMe_2_. The following order of the catalytic activity was found for the complexes bearing R_2_ = *t*Bu based on TOF calculations: **4a** (R_1_-X = CH_2_OMe) > **8a** (R_1_-X = 2-tetrahydrofuryl) > **1a** (R_1_-X = CH_2_NMe_2_) > **7a** (R_1_-X = 2-pyridyl).

The structure of the complexes was reflected in *M*_n_ values of the resulting polymers as well. The catalytic systems based on the complexes with *N*-donor groups provided the product characterized by the much higher molecular mass than the ones with *O*-donor pendant groups. Moreover, in the case of the series complexes with R_1_-X equal to CH_2_OMe or CH_2_NMe_2_ group, introducing F atoms as R_2_ substituents resulted in an increase in the molecular mass of the resulting polyester compared to the similar complexes having *t*Bu or MeO substituents. The increase was higher for the complex with R_1_-X = CH_2_NMe_2_.

The presence of F atoms in phenolate units was also reflected in the activity of the complexes with aliphatic pendant groups. However, it advantageously affected only the activity of the complexes with the *N*-donor pendant group, while the complex with *O*-donor group turned out to be less active compared to the similar complexes with R_2_ equal to H, *t*Bu, or MeO.

The most active of the examined complexes, complex **3a** (R_1_-X = CH_2_NMe_2_, R_2_ = F), was able to convert quantitatively (or nearly quantitatively) and selectively the series of six different epoxides (SO, ECH, PGE, 4-VCHO, PO, and BO) in the alternating ring-opening copolymerization with phthalic, using the following reaction conditions: the molar ratio [PA]_0_:[epoxide]_0_:[Cr]_0_:[DMAP] =250:250:1, 110 °C, toluene as a solvent, and reaction time of 240 min. The ^1^H NMR analysis proved that, in the performed ROCOP reactions, the obtained poly(alkylene phthalates) contained less than 1 mol% ether units in most cases. The resulting products were characterized by *M_n_* up to 20.6 kg/mol, a narrow distribution of molecular mass, and *T*_g_ ranged from 47.7 to 141.8 °C.

## Data Availability

The data presented in this study are available on request from the corresponding author.

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
