# Peer review of "Copolymerization of Phthalic Anhydride with Epoxides Catalyzed by Amine-Bis(Phenolate) Chromium(III) Complexes"

_polymers, 2021, doi:10.3390/polym13111785_

Round 1

Reviewer 1 Report

The paper of Bester and co-authors deals with chromium complexes as catalysts in the copolymerization of phthalic anhydride and epoxides. The activity of the catalysts is quite low, however a significant number of experiments exploring different parameters and conditions have been carried out. In all cases the authors try to explain the results also in reference to the literature. The paper is well written although in some cases the discussion is too long and could be reduced. Moreover, it would be useful for the readers to report in Table 1 also the data concerning the ROCOP in the presence of PPNCl and the PPNCl/Cr catalytic system. According to the reviewer the paper could be published in “Polymers” after minor revision.  

Author Response

RESPONSE TO REVIEWER REPORT:

1) The paper is well written although in some cases the discussion is too long and could be reduced.

According to reviewer suggestions, we shortened somewhat the Results and Discussion part by removing phrases which seem not to be so essential for the discussion of obtained results. We hope that it makes the manuscript more concise.

2). Moreover, it would be useful for the readers to report in Table 1 also the data concerning the ROCOP in the presence of PPNCl and the PPNCl/Cr catalytic system

We did not decide to add to the revised version of our manuscript the results showing the activity of PPNCl and the catalytic system composed of a amine-bis(phenolate) chromium(III) complex and PPNCl. It would enlarge the manuscript unnecessarily.

In our work, we concentrated only on examining the properties of nonionic basic cocatalysts. It followed from the results published previously, for instance by Duchateau et al. (Macromolecules 2012, 45, 1770−1776 (https://dx.doi.org/10.1021/ma2025804) and Hošt’álek et al. (Eur. Polym. J. 2017, 88, 433-447, https://doi.org/10.1016/j.eurpolymj.2017.01.002). As was showed by Duchateau et al., the systems based on salen metal complexes and PPNCl showed so high catalytic activity that it was difficult to find the differences in activity of the examined complexes. However, the use of nonionic bases by the authors revealed the differences in activity of the examined complexed clearly. Furthermore, Hošt’álek et al. showed that PPNCl used as a catalyst of the ROCOP reactions of cyclic anhydrides and epoxides allowed to obtain relatively high conversions of substrates under the similar condition as was applied in our study.

In order not to draw attention of a reader on the potential possibility of the use of PPNCl a co-catalyst, we decided to remove from the revised version of the manuscript the following fragment: “Taking into account this fact, being concerned that PPNCl could mask the actual influence of the structure of the amine-bis(phenolate) chromium(III) complexes on their catalytic activity, only nonionic compounds were used in our study as much less active catalysts/co-catalysts for the ROCOP reaction compared to PPNCl [11,12].” We hope that this change does not influence negatively on the quality of the manuscript. The removal of the mentioned fragment gave us the possibility of further shortening the discussion.

Furthermore, according to your suggestion we did the best to improve spell mistakes performing additional linguistic correction of our manuscript as well. The revised version of our manuscript is uploaded as an attachment.

Reviewer 2 Report

This article is devoted to the catalytic copolymerization of phthalic anhydride with various epoxides. This research has been carried out at a high level, and the results obtained are new and original and have great practical and scientific significance. I recommend accepting the article in its current form.

Author Response

We would like to thank the reviewer for the positive reception of our new manuscript. According to your suggestions we did the best to improve spell mistakes performing additional linguistic correction of our manuscript as well.